# Recurrent Sinonasal Squamous Cell Carcinoma: Current Insights and Treatment Advances

**DOI:** 10.3390/cancers17010004

**Published:** 2024-12-24

**Authors:** Meryl B. Kravitz, Vivek Annadata, Benjamin Ilyaev, Charles C. L. Tong, Judd H. Fastenburg, Mark B. Chaskes

**Affiliations:** 1Department of Otolaryngology/Head & Neck Surgery, Zucker School of Medicine, Hofstra University, New York, NY 11040, USA; ctong1@northwell.edu (C.C.L.T.); jfastenburg@northwell.edu (J.H.F.); mchaskes@northwell.edu (M.B.C.); 2Department of Otolaryngology/Head & Neck Surgery, Montefiore Medical Center, New York, NY 10467, USA; 3Donald and Barbara Zucker School of Medicine, Hofstra University, New York, NY 11549, USA; vannadata1@pride.hofstra.edu (V.A.); bilyaev1@pride.hofstra.edu (B.I.)

**Keywords:** squamous cell carcinoma, tumor, recurrence, endoscopic, open, sinonasal

## Abstract

Squamous cell carcinoma (SCC) is the most common cancer affecting the sinonasal tract. Unfortunately, it has a high rate of recurrence, affecting nearly half of patients who are initially treated. Nonetheless, there are limited guidelines for diagnosing and managing recurrent disease. This review discusses diagnosis, prognostic factors, and available treatment algorithms for recurrent cases. For eligible patients, surgery remains the most effective long-term treatment. When surgery is not an option, chemotherapy and radiation are used either to help control the disease or to ease symptoms. Newer treatments, such as immunotherapy and advanced radiation techniques, are being actively researched as promising alternatives.

## 1. Introduction

Squamous cell carcinoma (SNSCC) is the most common malignancy affecting the sinonasal tract. Despite recent advancements in treatment, still, nearly half of patients will suffer from recurrent disease [1,2,3,4,5,6,7]. Unfortunately, given the overall rarity of SNSCC, and recurrent SNSCC (rSNSCC), specifically, evidence is sparse regarding the management of recurrent disease. This review highlights recent advancements and contemporary evidence guiding the detection and treatment of rSNSCC. An algorithm for management of rSNSCC is proposed (Figure 1).

## 2. Surveillance and Detection of Recurrence

Given the high rate of recurrence, post-treatment surveillance is a critical facet of patient care [2,5,8,9]. Despite this, there are no accepted universal guidelines. While clinicians may follow the National Comprehensive Cancer Network (NCCN) guidelines for head and neck cancer, these protocols may be inappropriate for sinonasal malignancies, which often present at more advanced stages than head and neck malignancies due to their occult nature and more often recur in a more delayed fashion [10,11]. While most recurrences of head and neck SCC will occur within 1 year of initial treatment, several studies have found median time to recurrence for SNSCC to be closer to 1.5 years [3,7,8,9,11,12,13].

Endoscopic examination is the workhorse of postoperative surveillance given its accessibility and cost effectiveness. Most commonly, clinicians follow the NCCN guideline examination protocol for head and neck cancer: every 1–3 months during the first year, every 2–6 months during the second year, every 4–8 months until 5 years post-treatment, and then annually thereafter [8]. Although diagnosis of recurrence by nasal endoscopy is highly specific, this method may only diagnose a minority of rSNSCC (estimated 10–37.6%) due to limitations visualizing the full extent of tumor invasion [8,11,14]. The positive predictive value of endoscopy as the modality to diagnose SNSCC recurrence increases when other suspicious symptoms are reported (83% vs. 13% in asymptomatic patients) [10].

Imaging is an important adjunct to endoscopy and allows for detection of submucosal or poorly visualized recurrences. However, ambiguous post-treatment changes detected on imaging may be vulnerable to a higher rate of false positives. Consequently, imaging overall has a high negative predictive value (>90%) for diagnosing recurrence and can provide reassurance during routine surveillance [10,11]. Computed tomography (CT), magnetic resonance imaging (MRI), and positron emission tomography-computed tomography (PET-CT) have all been described as modalities for imaging surveillance. Nonetheless, there is no established guideline for imaging surveillance.

Although CT is not the best modality for detecting early sinonasal recurrences, it is frequently employed due to advantages in cost and accessibility [10]. MRI, however, is often regarded as the “gold standard” to detect recurrence of sinonasal malignancies [10]. Compared to CT or PET-CT, MRI may have a higher positive predictive value for detecting recurrence [15]. Many clinicians follow NCCN head and neck guidelines which recommend a baseline MRI at 3–4 months after treatment. High resolution of soft tissue and demonstration of potential extension into critical adjacent structures such as the brain, orbit, and neurovascular structures make this modality highly effective [15,16]. Dynamic contrast enhanced MRI and diffusion weighted MRI can assist in differentiating radiation-related changes from recurrence. Nonetheless, MRI findings are highly interpreter-dependent, and differences in readings may lead to unnecessary biopsies. PET-CT is also often used in the imaging surveillance of post-treatment SNSCC. While NCCN guidelines recommend PET-CT 3 months post-treatment in head and neck cancers, the optimal timing for sinonasal malignancies is unclear. PET-CT in the early post-treatment period for sinonasal malignancies has demonstrated a low specificity and positive predictive value in detecting recurrence (40 and 50%, respectively) [10], due to a prolonged hypermetabolic state of the sinonasal region, persisting up to 1 year after treatment [17]. Gil et al. found improved utility of PET-CT over 6 months post-treatment, with a positive predictive value of 66% at this time point [18]. Furthermore, PET-CT may be particularly useful in identifying distant metastases, affecting up to 10% of patients with rSNSCC [2,3,10,17,19]. As such, some advocate to alternate MRI and PET-CT every 3 months for the first year post-treatment, followed by alternating every 6 months until four years post-treatment [11,20].

## 3. Local Recurrences

Local recurrence is the primary mode of treatment failure, occurring in up to 50% of patients [1,2,4,5,8]. Many factors, including size, T-stage, and histologic features such as perineural or lymphovascular invasion of the primary tumor have been implicated as risk factors of local recurrence [1,2,4,21,22]. Location may also play a role in local recurrence. The maxillary sinus subsite has been associated with higher rates of local recurrence when compared to other subsites of the sinonasal tract [2,6,13,23]. It is likely that these high-risk characteristics are surrogates for more locally advanced disease at time of diagnosis, more aggressive tumor biology, or more complex resections and therefore, poorer margin management at the time of initial resection.

Adjuvant radiation after primary surgery for SNSCC is associated with decreased risk of locoregional recurrence compared to surgery alone [24]. This is the justification for adjuvant radiation as the standard of care for most SNSCC. Nonetheless, patients who have unresectable tumors and receive primary radiation may have a higher rate of in-field recurrence, related to preservation attempts of adjacent neurovascular structures [8].

## 4. Regional Recurrences

Nodal failure occurs in the minority of recurrences and is more often accompanied by local recurrence than as isolated nodal failure. Nonetheless, SNSCC has a greater propensity for regional recurrence than other sinonasal cancers. Large database studies have found up to 12.5% of patients with occult nodal neck disease at diagnosis, and up to 20% with regional recurrences without elective neck treatment (radiation or surgery) [25,26,27,28,29]. Despite this, elective nodal treatment for primary SNSCC is not universally performed and varies according to T-stage, subsite, histology, and provider preference [28,30]. Therefore, it is plausible that some neck disease classified as regional recurrences may in fact be persistence.

Risk factors for regional recurrences include history of nodal disease, prior neck dissection or radiation, positive surgical margins, site of primary tumor, and higher T-stage [27,31]. Lymphatic drainage pathways play a large role in regional recurrence. Cantù et al. observed that patients with maxillary sinus malignancies had higher risk of lymph node recurrences compared to ethmoid sinus malignancies due to tumor involvement of the floor of the maxillary sinus and nasal cavity, with invasion to areas with stronger lymphatic drainage patterns, including mucosa of the hard palate or upper gingiva [27,32,33]. Similarly, in a retrospective review of 48 patients with SNSCC, Janik et al. found that the septal subsite was associated with higher rates of lymph node metastasis compared to those from the nasoethmoidal complex (45.5% vs. 2.8%) [34]. Again, rich lymphatic drainage to the nose and upper lip promotes early metastasis from the anterior septum.

## 5. Salvage Surgery

Salvage surgery is defined as a second attempt at curative treatment after recurrence. Salvage surgery is the mainstay of treatment for resectable tumors, with or without adjuvant radiotherapy or chemotherapy [8,35]. Salvage procedures have been found to improve OS and DFS compared to other modalities of treatment for recurrent disease. Curative salvage surgery can offer sustainable cure with similar survival rates as primary surgery (Table 1) [2,5,9,36].

Unfortunately, it is estimated that salvage surgery may only be available to approximately 70% of patients with rSNSCC [8,35]. Many factors may influence patients’ eligibility for salvage surgery including proximity to neurovascular structures, ability to achieve a R0 resection, reconstruction considerations, presence of metastatic disease, and patients’ fitness for surgery.

A retrospective study of 79 cases of rSNSCC found that time to recurrence may be predictive of eligibility for salvage surgery. Longer disease-free interval (DFI) was associated with higher probability of eligibility for salvage [2]. A similar finding was found with recurrent oral cavity cancer, and authors attributed long recurrence intervals to less aggressive cancers [37]. Another hypothesis suggests that late recurrences may form from few cancer cells which are wrapped in fibrotic post irradiation scar tissue, inhibiting growth of the recurrent tumor and resulting in more feasible salvage resection [38]. Mode of diagnosis may also predict candidacy for surgery. Tzelnick et al. found that patients with recurrences diagnosed by nasal endoscopy were more likely to be candidates for salvage surgery, compared to those diagnosed by imaging alone [8]. This finding may be a proxy for resectability, whereby lesions that are less visible are also less resectable.

The presence of positive margins is a key cause for failure after salvage surgery. A retrospective study of 76 cases of salvage surgery for recurrent maxillary cancer, 92% of which were SCC, found that positive surgical margins decreased 2-year local control rate (14.3% vs. 83.3%) and 2-year OS (14.3% vs. 66.7%). Furthermore, patients who underwent salvage surgery with positive margins did not live significantly longer than those who did not undergo salvage surgery (8.3 vs. 4.0 months). Thus, the authors recommend that if surgical margins cannot likely be achieved, salvage surgery should not be performed as it may exclude the patient from alternative treatment modalities [36].

As such, primary site, grade, and tumor extension may predict success of surgical salvage due to anatomic constraints at achieving negative margins. Nasal cavity and ethmoid sites have shown improved OS and DFI after salvage surgery compared to maxillary and frontal sites [9,39]. One hypothesis described is that advantages are due to favorable surgical location, where surgical excision of the skull base involvement is feasible [9].

Similarly, recurrence extending to critical skull base structures as well as the orbit has been found to be negative predictors for successful salvage [9]. In a case-series of 42 patients undergoing surgical salvage, post-salvage surgery survival was negatively associated with tumor extension into carotid artery, perineural space, and clivus [9]. The authors attribute this to inability to obtain negative margins due to these structures. Orbital invasion was also negatively associated with OS and DFI, even if orbital exenteration was performed. Authors hypothesize that orbital invasion may be a surrogate for more aggressive tumors. Interestingly, intracranial invasion was not a predictor of failed salvage, and adequate resection was not impacted by involvement of the anterior skull base. Importantly, there was no subgroup analysis for SCC, representing 45% of the cohort [9].

Reconstructive options are often critical to consider prior to performing salvage surgery. Reconstruction often requires local pedicelled mucoperiosteal flaps, multilayer grafts, and synthetic grafts. However, as traditional reconstruction options are often not available for use either secondary to prior treatment or tumor involvement, regional pedicled flaps and soft tissue free flaps may be necessary. In particular, vascularized local, regional, or free flaps should be considered for defects that are large, posterior or lateral, or in patients with vascular concerns due to prior RT, planned irradiation, or osteonecrosis.

**Table 1 cancers-17-00004-t001:** Survival Outcomes in Recurrent Sinonasal Squamous Cell Carcinoma (rSNSCC).

Study	Number of Patients w/ Recurrent Sinonasal Cancer (% SCC)	Median Disease-Free Interval	% Patients Receiving Completing Salvage Surgery	OS After Recurrence
Kaplan et al., 2016 [9]	42 (45%)	16.6 months (SCC only)	100%	42.1% 5-year OS (SCC only)
Ono et al., 2019 [36]	24 * (92%)	NA	58%	48.5% 2-year OS after salvage surgery 11.1% 2-year OS after no salvage surgery
Orlandi et al., 2020 [3]	44 (34%)	14.8 months (SCC only)	44%	29.5 months median OS after salvage 4.6 months median OS after palliative chemotherapy 0.9 months median OS after supportive care
Lehrich et al., 2021 [39]	207 * (60.4%)	NA	100%	67% 2-year OS (SCC only) 47% 5-year OS (SCC only)
Kacorzyk et al., 2022 [2]	24 (100%)	NA	62.50%	118.5 months median OS after salvage 10 months median OS without effective salvage
Mattavelli et al., 2022 [13]	118 (25.4%)	18 months	79.50%	71.7% 2-year OS 42.5% 5-year OS
Tzelnick et al., 2022 [8]	24 (37.5%)	26.5 months (SCC only)	70% (SCC only)	60.9% 5-year OS
Contrera et al., 2024 [40]	11 (100%)	NA	100%	15.8 months median OS 45% 2-year OS 36% 5-year OS

A table with recently published studies of survival outcomes in patients with rSNSCC. Data represent the entire cohort unless specified as SCC only. OS: overall survival. * Study included persistence and recurrence in cohort.

## 6. Chemotherapy and Radiation Therapy

Salvage surgery is the only contemporary modality offering possible sustainable cure for rSNSCC. When salvage surgery cannot be performed, the treatment paradigm shifts from curative treatment to life-prolonging treatment, including chemotherapy and radiation [9]. As such, many patients will consider palliative approaches to reduce symptom burden and improve quality of life. Kaplan et al. advocate that patients with recurrent sinonasal malignancies of poor grade (including SCC) from a non-ethmoid location with skull base or orbital invasion should be considered for palliative regimen given poor prognosis even with life-prolonging treatments. Nonetheless, radiation and chemotherapy continue to be used as life-prolonging treatments and may improve survival for some patients with recurrences. Systemic chemotherapy is often employed in management of rSNSCC as a palliative treatment or in conjunction with salvage radiation or surgery. Systemic regimens often include platinum, anthracycline, taxane, and alkylating agents. In a review of 44 patients with recurrent sinonasal malignancies, 42% of which were SCC, palliative chemotherapy was found to increase OS in about 50% patients who achieved objective response (29.2 months vs. 4.4 months) [3].

Re-irradiation (re-RT) may be offered to patients who are not eligible for salvage surgery or as an adjuvant treatment to salvage surgery, though the benefits of this modality are not well described [1]. A majority of patients with primary SNSCC will receive RT as a radical or adjuvant treatment for primary disease [1,5]. Although re-RT may prolong OS, severe toxicity leading to high morbidity and mortality often limits its use. Due to proximity of vital neurovascular structures, re-RT increases risk of catastrophic neurovascular events. The decision to pursue irradiation may be based on tissue tolerability, dose constraints, anatomical subsite, efficacy of first radiotherapy, curative or palliative intent, access to photon versus charged-particle therapy, toxicity management, and patient’s goals and expectations.

Sustainable cure is less likely in patients pursuing primary re-RT. As these tumors are universally highly advanced precluding salvage surgical treatment, RT is often limited due to toxicity and anatomy constraints. Nonetheless, re-RT with systemic therapy may prolong survival compared to systemic therapy alone. Chen et al. found that for locoregionally recurrent head and neck cancer, radiation in addition to chemotherapy improved 2-year OS (32% vs. 11%) and progression free survival (31% vs. 7%), although new severe toxicity was higher in patients who received radiation (53% vs. 28%) [41]. Importantly, this study did not include sinonasal malignancies in their analysis.

A similar 2-year OS as the forementioned study was found in patients receiving re-RT alone for rSNSCC. Yamazaki found a 2-year OS of 35%, median survival time of 16.3 months for patients re-irradiated with advanced photon radiotherapy for SNSCC [42]. Female sex and gross tumor volume less than 25 cm^3^ were positive prognostic factors for longer survival. Interestingly, there was no correlation with prognosis and age, prior surgery, chemotherapy, interval between radiotherapy treatments, prescribed dose and skull base/neurovascular involvement. Severe toxicity was reported by 22% of patients, with the most common complications being visual impairment, hemorrhage, and fistula formation.

Particle therapy may offer higher survival than conventional photon modalities. A systematic review comparing radiation algorithms for recurrent head and neck cancer (including sinonasal malignancies) found higher 2-year OS rates with particle therapy (including proton and carbon) compared to advanced photon radiotherapy, including stereotactic body radiation therapy (SBRT) and Intensity-Modulated Radiation Therapy (IMRT) (Proton 33–80% and Carbon 59–82% vs. SBRT 14–58% and IMRT 12–68%) [43]. Importantly, the prevalence of severe toxicity was less in SBRT relative to other modalities (Proton 0–33%, Carbon 0–37%, SBRT 0–18%, IMRT 15–48%).

## 7. Management of the Neck

Management of nodal recurrence depends on the presence and extent of local recurrence, history of neck dissection/radiation, and bulk of lymphadenopathy. Cervical relapses are frequently accompanied by uncontrollable primary or distant disease. As such, there is often no added benefit to addressing nodal recurrence [44]. Given that local disease progression is the primary cause of failure at any stage, therapy to achieve maximum local control is considered more important than elective neck treatment.

Nonetheless, neck dissections are indicated in certain scenarios. Type of neck dissection (selective, comprehensive, or radical) varies in studies without evidence to support a superior method [1,45]. Patients with isolated lymph node recurrences with stable or absent local disease may be good candidates for neck dissection, although this is rare, occurring in less than 10% of patients after treatment of primary site [44]. In a retrospective study of 399 patients treated for maxillary sinus cancer, 39% being SCC, 31 developed isolated lymph node recurrence, 30 of which underwent salvage neck dissection with or without adjuvant therapy [25]. Only 2 of the 31 (6.5%) patients died from nodal-only metastasis during follow up.

Management of regional recurrences that accompany local disease may vary depending on status and treatment plan for local disease, and there is no consensus in the literature. In cases with resectable local recurrences, neck dissection may be pursued at the time of surgery, especially for those requiring free tissue transfer for reconstruction and cervical recipient vessel dissection [1,30,44].

In non-resectable recurrences, re-irradiation of local and regional disease may be considered to improve survival time and palliation, although with low likelihood for sustainable cure [3,42]. A meta-analysis of survival rate with re-irradiation for sinonasal malignancies found that lymph node involvement was associated with decreased 2-year OS (26% vs. 42.7% in lymph node negative cases) [42]. Palliative chemotherapy is often offered at this juncture.

## 8. Future Directions

Tumor surveillance remains an active area of research given the limitations of endoscopy and prevalence of false positive findings on surveillance imaging. A study by Lin et al. developed and tested a nomogram based on a deep learning MRI radiomics model and clinicopathological qualities to predict early recurrence in SNSCC [46]. The model performed better than clinical or radiomics models alone [46]. In coming years, radiomics-based machine learning models may become increasingly utilized to improve detection of recurrences during post-treatment surveillance [46,47].

Additionally, new treatment options and treatment delivery methods are areas of ongoing investigation as they pertain to rSNSCC. The use of systemic immunotherapy, such as nivolumab or pembrolizumab, has been probed as an additional treatment for recurrent and metastatic SNSCC [1]. These new medications have been approved for platinum-resistant recurrent or metastatic HNSCC with PD-L1 positivity, although landmark trials excluded sinonasal sites. Nonetheless, Riobello et al. found that 26% patients of SNSCC had greater than 50% PD-L1 expression, suggesting that PD-L1 may be an effective immunotherapeutic target for these patients [48]. Small studies have found prolonged survival and partial or complete response in certain patients, although larger scale studies are ongoing [49,50]. Furthermore, the employment of intra-arterial chemotherapy has also been studied in locally advanced SNSCC, although its use in recurrence remains an area of future investigation [36,51].

Finally, nanotherapy is a growing avenue of research with the goal of optimizing radiation, chemotherapy, and immunotherapy effectiveness within the tumor micro-environment [52,53,54]. Several nanoimmunotherapeutics are under investigation as delivery platforms for use in head and neck SCC [54]. However, to date, there are no studies evaluating the role of nanotherapy as it specifically relates to rSNSCC.

## 9. Conclusions

SNSCC is the most common malignancy of the sinonasal tract. Unfortunately, locoregional recurrence occurs in nearly half of patients with SNSCC. Restrictions in endoscopic visualization and variability in interpretation and high false positives in surveillance imaging limit detection of recurrence. Sinonasal specific surveillance protocols are needed to better address the unique features of this disease, including prolonged hypermetabolic states, compared to other head and neck SCC. Current data suggest that salvage surgical resection is the only durable curative approach to treatment of these patients. However, with emerging treatment options for SCC of other subsites in the head and neck, including immunotherapy and novel delivery methods for current agents, the management of rSNSCC remains an area of ongoing investigation.

## Figures and Tables

**Figure 1 cancers-17-00004-f001:**
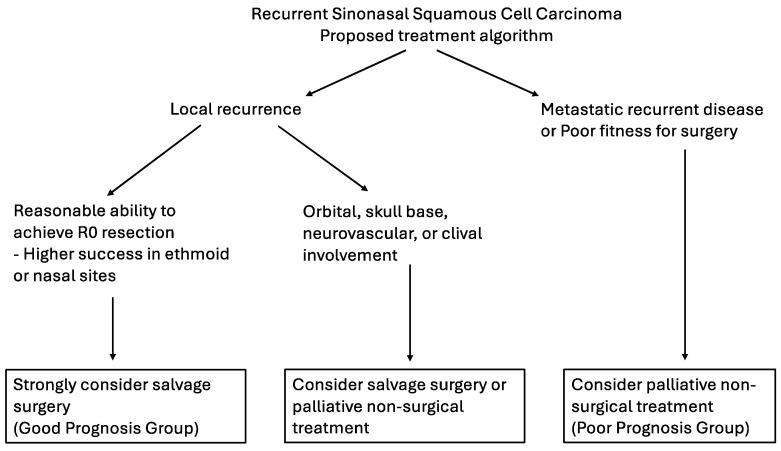
Recurrent sinonasal squamous cell carcinoma—proposed treatment algorithm.

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
