# Peer review of "Recurrent Sinonasal Squamous Cell Carcinoma: Current Insights and Treatment Advances"

_cancers, 2024, doi:10.3390/cancers17010004_

Round 1
Reviewer 1 Report
Comments and Suggestions for Authors
Squamous cell carcinoma (SNSCC) was reviewed, highligting detection and main treatment approaches. This is a short review, however it is comprehensive. I think it might be interesting for the researchers since there is not much review on this topic. The manuscript is well written. But, there is no Figures and table to help the compreghension of the content. I suggest that some of the major detection and therapy strategies can be explianed by using some figures from the citations.
Author Response
Comments 1: Squamous cell carcinoma (SNSCC) was reviewed, highligting detection and main treatment approaches. This is a short review, however it is comprehensive. I think it might be interesting for the researchers since there is not much review on this topic. The manuscript is well written. But, there is no Figures and table to help the compreghension of the content. I suggest that some of the major detection and therapy strategies can be explianed by using some figures from the citations.
Response 1: Thank you for your response. We have included a figure and a table in the new manuscript (Page 2 and page 6)
Reviewer 2 Report
Comments and Suggestions for Authors
The authors have written a review on etiology, pathophysiology, diagnosis, prognostic factors, and treatment modalities of recurrent Squamous cell carcinoma (rSNSCC). The review is brief and encompasses the topics of surveillance and recurrence detection, salvage surgery, chemotherapy and radiation therapy and management of neck. I have the following comments:
The review ends abruptly. Please elaborate the discussion, include future perspective and the author's view.
Please include a topic on nanotechnology based treatment modalities for rSNSCC.
There are no figures and tables present in the article. Please include at least two representative figures and one table to make the readers understand the topic in a better way.
There are some typos that should be corrected. One is mentioned below but many more can be present. So, check the entire manuscript for spelling and grammar.
Line # 168: "Kaplan et Al" will be "Kaplan et al"
Some recent studies should be included in the manuscript (doi: 10.7150/ijbs.47068, https://doi.org/10.1016/j.bcab.2024.103109)
I recommend a major revision
Author Response
Comment1:
The review ends abruptly. Please elaborate the discussion, include future perspective and the author's view.
Response 1: Thank you for your thoughtful response. We have included these topics on pages 7 and 8 in our revised manuscript.
Comments 2: Please include a topic on nanotechnology based treatment modalities for rSNSCC.
Response 2: Thank you for pointing this out. We agree with this comment. We have included these topics on pages 7 and 8 in our revised manuscript.
Comment 3: There are no figures and tables present in the article. Please include at least two representative figures and one table to make the readers understand the topic in a better way.
Response 3: Thank you for your response. We have included a figure and a table in the revised manuscript (Page 2 and page 6).
Comment 4: There are some typos that should be corrected. One is mentioned below but many more can be present. So, check the entire manuscript for spelling and grammar. Line # 168: "Kaplan et Al" will be "Kaplan et al"
Response 4: Thank you for pointing this out. We have fixed this and any other grammatical errors.
Comment 5: Some recent studies should be included in the manuscript (doi: 10.7150/ijbs.47068, https://doi.org/10.1016/j.bcab.2024.103109)
Response 5: Thank you for pointing these out, we have included these two studies in our future directions section (Pages 7 and 8).
Reviewer 3 Report
Comments and Suggestions for Authors
In the review manuscript “Recurrent Sinonasal Squamous Cell Carcinoma: Current Insights and Treatment Advances” Kravitz et al., provide an overview of the recurrence of squamous cell carcinoma of the sinonasal tract. The review is up to date and provides a critical evaluation of the presented material, especially in respect to imaging methodologies.
Weakness: absence of illustrations for the stronger points made, or where an illustration would help.
Possible mistakes in the spelling of names (Judd H. Fastenburg might need to be corrected to Judd H. Fastenberg).
Optional:
Discuss machine learning. There are some papers available. One example is,
MRI radiomics-based machine learning model integrated with clinic-radiological features for preoperative differentiation of sinonasal inverted papilloma and malignant sinonasal tumors
https://pubmed.ncbi.nlm.nih.gov/36212455/
https://pmc.ncbi.nlm.nih.gov/articles/PMC9538572/
Author Response
Comment 1: Weakness: absence of illustrations for the stronger points made, or where an illustration would help.
Response 1 : Thank you for your suggestion, we agree. We have included these two studies in our future directions section (Pages 7 and 8).
Comment 2: Possible mistakes in the spelling of names (Judd H. Fastenburg might need to be corrected to Judd H. Fastenberg).
Response 2: Thank you for pointing this out. We have fixed this and any other grammatical errors.
Comment 3: Discuss machine learning. There are some papers available. One example is:
MRI radiomics-based machine learning model integrated with clinic-radiological features for preoperative differentiation of sinonasal inverted papilloma and malignant sinonasal tumors
https://pubmed.ncbi.nlm.nih.gov/36212455/
https://pmc.ncbi.nlm.nih.gov/articles/PMC9538572/
Response 3: Thank you for pointing these out, we have included these two studies in our future directions section (Pages 7 and 8).
Round 2
Reviewer 2 Report
Comments and Suggestions for Authors
The authors have addressed most of the queries raised by me, but a few important points have been missed out.
Comment 3: There are no figures and tables present in the article. Please include at least two representative figures and one table to make the readers understand the topic in a better way.
Response 3: Thank you for your response. We have included a figure and a table in the revised manuscript (Page 2 and page 6).
The table that is included mentions the author's works that has already been mentioned in the text. For example the work of Kalpan et al. This creates redundancy in the manuscript. Please remove from any one place, either from the text or from the table.
The authors have mentioned that they have included a figure, but the Figure is not like a figure. It is a summary or highlights about the topic. There is no motif or image in the figure. Please re do the figure or insert a separate figure. Cancers is a flagship journal of MDPI and it publishes article of high standards. Thus, a high quality figure is necessary.
Comment 5: Some recent studies should be included in the manuscript (doi: 10.7150/ijbs.47068, https://doi.org/10.1016/j.bcab.2024.103109)
Response 5: Thank you for pointing these out, we have included these two studies in our future directions section (Pages 7 and 8).
In this response the authors claim to cite both the articles, but I found that they cited only one. Nanoformulations used for cancer treatment is the need of the hour and must be mentioned in any recent review on cancer. Please include.
I found spelling mistakes again. Please be careful while preparing the revision.
Line #293: ".......increasingly utilizd .....". Correct the spelling of "utilized".
I recommend a major revision.
Author Response
Dear reviewer,
We appreciate the time and effort that you dedicated to providing feedback on our manuscript and are grateful for the insightful comments on and valuable improvements to our paper.
Comment 1: The table that is included mentions the author's works that have already been mentioned in the text. For example the work of Kaplan et al. This creates redundancy in the manuscript. Please remove from any one place, either from the text or from the table. The authors have mentioned that they have included a figure, but the Figure is not like a figure. It is a summary or highlights about the topic. There is no motif or image in the figure. Please re do the figure or insert a separate figure. Cancers is a flagship journal of MDPI and it publishes article of high standards. Thus, a high quality figure is necessary.
Response 1: Thank you for your feedback. We have removed any redundant text that overlaps with the table. We have also included a new figure that outlines a proposed treatment algorithm based on reviewed articles. We believe this is a high-quality figure that can guide practice for rhinologists.
Comment 2: In this response the authors claim to cite both the articles, but I found that they cited only one. Nanoformulations used for cancer treatment is the need of the hour and must be mentioned in any recent review on cancer. Please include. (doi: 10.7150/ijbs.47068, https://doi.org/10.1016/j.bcab.2024.103109)
Response 2: Thank you for this feedback. We have cited this article.
Comment 3: I found spelling mistakes again. Please be careful while preparing the revision. Line #293: ".......increasingly utilizd .....". Correct the spelling of "utilized". Response 3: Thank you for detecting this error. All spelling errors are now fixed.
Comment 3:Thank you for detecting this error. All spelling errors are now fixed.
Round 3
Reviewer 2 Report
Comments and Suggestions for Authors
The authors have addressed all the queries raised by honorable reviewers and the manuscript is now much improved. I accept the article in its present form for publication.